

# Minimum infusion rate and adrenocortical function after continuous infusion of the novel etomidate analog ET-26-HCl in rats

Junli Jiang[1,2,*], Bin Wang[1,*], Zhaoqiong Zhu[2], Jun Yang[1], Jin Liu[1,3] and Wensheng Zhang[1,3]

[1] Laboratory of Anesthesia and Critical Care Medicine & Translational Neuroscience Centre, West China Hospital, Sichuan University, Chengdu, Sichuan, China
[2] Department of Anesthesiology, Affiliated Hospital of Zunyi Medical College, Zunyi, Guizhou, China
[3] Department of Anesthesiology, West China Hospital, Chengdu, Sichuan, China
[*] These authors contributed equally to this work.

Corresponding author
Wensheng Zhang,
zhang_ws@scu.edu.cn

## ABSTRACT

**Background.** Because etomidate induces prolonged adrenal suppression, even following a single bolus, its use as an infused anesthetic is limited. Our previous study indicated that a single administration of the novel etomidate analog methoxyethyletomidate hydrochloride (ET-26-HCl) shows little suppression of adrenocortical function. The aims of the present study were to (1) determine the minimum infusion rate of ET-26-HCl and compare it with those for etomidate and cyclopropyl-methoxycarbonylmetomidate (CPMM), a rapidly metabolized etomidate analog that is currently in clinical trials and (2) to evaluate adrenocortical function after a continuous infusion of ET-26-HCl as part of a broader study investigating whether this etomidate analog is suitable for long infusion in the maintenance of anesthesia.

**Method.** The up-and-down method was used to determine the minimum infusion rates for ET-26-HCl, etomidate and CPMM. Sprague-Dawley rats ($n = 32$) were then randomly divided into four groups: etomidate, ET-26-HCl, CPMM, and vehicle control. Rats in each group were infused for 60 min with one of the drugs at its predetermined minimum infusion rate. Blood samples were drawn initially and then every 30 min after drug infusion to determine the adrenocorticotropic hormone-stimulated concentration of serum corticosterone as a measure of adrenocortical function.

**Results.** The minimum infusion rates for etomidate, ET-26-HCl and CPMM were 0.29, 0.62, and 0.95 mg/kg/min, respectively. Compared with controls, etomidate decreased serum corticosterone, as expected, whereas serum corticosterone concentrations following infusion with the etomidate analogs ET-26-HCl or CPMM were not significantly different from those in the control group.

**Conclusion.** The corticosterone concentrations tended to be reduced for the first hour following ET-26-HCl infusion (as compared to vehicle infusion); however, this reduction did not reach statistical significance. Thus, further studies are warranted examining the practicability of using ET-26-HCl as an infused anesthetic.

## INTRODUCTION

Etomidate, one of the most frequently used intravenous anesthetics, has many favorable properties, such as its ability to maintain hemodynamic stability and generate low incidences of respiratory depression and anaphylaxis. However, the adrenocortical insufficiency caused by etomidate restricts its clinical applications. Several studies have shown that as little as a single administration of etomidate may induce adrenocortical insufficiency, and the increased propensity for this may last 48 h after the administration of etomidate (*Hildreth et al., 2008*; *Tekwani et al., 2008*).

In the 1980s, etomidate was infused in a continuous manner in critical patients to maintain sedation; however, studies later showed that this dosage regimen increased mortality (*Ledingham & Watt, 1983*; *Watt & Ledingham, 1984*). Pharmacologists have spent years attempting to develop new etomidate analogs in their search for a drug that retains the desirable properties of etomidate but does not cause adrenocortical insufficiency. We recently reported on ET-26-HCl, a promising compound selected from dozens of other etomidate analogs that were designed using our synthesis strategy, showing that ET-26-HCl effectively produces reversible anesthesia, and that a signal administration does not significantly decrease plasma corticosterone levelsin beagle dogs (*Yang et al., 2017*). The primary aim of the present study was to evaluate the effect of ET-26-HCl on adrenal function after a continuous infusion.

## MATERIALS AND METHODS

### Animals and materials

All animal protocols used in the present study were approved by the Ethics Committee of the West China Hospital, Sichuan University, China (ethics approval No. 2015015A; date: 28/12/2012). Sprague Dawley rats weighing 225–350 grams were purchased from Chengdu Dassy Biological Technology Co. Ltd. (Chengdu, China) and cared for in accordance with the the Canadian Council on Animal Care's *Guide to the Care and Use of Experimental Animals* (Vol. 1 2nd ed., 1993). Five animals per cage were housed under standard conditions at a temperature of 22 °C and a humidity of 60% and with standard laboratory rat chow and water. The animals were allowed to acclimatize for 1 week.

Etomidate (2 mg/mL) formulated as an emulsion was purchased from B. Braun Melsungen AG, and propofol (10 mg/mL) formulated as an emulsion was purchased from AstraZeneca. The Laboratory of Anesthesia and Critical Care Medicine (West China Hospital, Sichuan University, China) synthesized ET-26-HCl using a previously published approach; ET-26-HCl was formulated as an aqueous solution (10 mg/mL) with 35% propylene glycol and then diluted with normal saline (0.9%; Kelun Pharmaceutical Co., Ltd.) to 6 mg/mL. Cyclopropyl-methoxycarbonylmetomidate (CPMM) was also

synthesized by our laboratory according to the issued patent (patent No., US9156825B2) and formulated as an aqueous solution (8 mg/mL) with 20% sulfobutylether-β-cyclodextrin (*Campagna et al., 2014*).

An infusion pump (Sino Medical-device Technology Co., Ltd.) was used to administer the sedatives and hypnotics, and a homoeothermic blanket was used to maintain the body temperature of the rats at 36–38 °C while they were anesthetized. A pulse oximeter placed on the upper right hind leg was used for monitoring oxygen saturation. Heart rate and rhythm and the respiratory rate were monitored with a BL-420S biological data acquisition and analysis system (Taimeng Software Co. Ltd., Chengdu, China). Oxygen (100%) was delivered at a rate of 2 L/min to each anesthetized rat through a face mask connected to a coaxial circuit.

## Determination of the minimum infusion rate

Rats ($n = 60$) were randomly assigned to three groups that received either etomidate, ET-26-HCl, or CPMM.

A 22-gauge catheter was inserted into the caudal vein of the rat for drug infusion. The minimum infusion rate (MIR) for each anesthetic was determined as previously described (*Ge et al., 2012*). The initial infusion rates of etomidate and CPMM were based on those that had been determined in previous studies (*Ge et al., 2013*). Based on our preliminary experimental results indicating that ET-26-HCl was one-third as potent a hypnotic as etomidate, the initial infusion rate of ET-26-HCl was three times greater than that of etomidate. The first subject in each group was administered the selected initial continuous infusion rate for one drug for 40 min. Then the animal's response to noxious stimulation was determined. A painful stimulus was provided by clamping the tail with an alligator clip and rolling the clamp at 1–2 Hz for 60 s or until the rat exhibited a purposeful response. Based on the presence or absence of purposeful movements of the extremities, the infusion rate for that drug was increased or decreased by 10% (*Li et al., 2012*) in the next rat and held constant for 40 min before the tail-clamping stimulation was repeated. The response of the rats to the painful stimulus was judged to be either negative or positive. When the rat showed a gross purposeful movement (e.g., limbs retracted or head twisted), the response would be considered positive, and the infusion rate of the drug for the next rat was increased. Conversely, when rats made no gross purposeful movement, the infusion rate for the next rat was reduced. A change in the direction of the response from negative to positive or positive to negative was defined as a pair, and the stimulation was repeated at different infusion rates until five pairs of responses were recorded. The MIR was calculated as the average of these five mean values.

## Evaluation of adrenocortical function

The infusion protocol for each rat began at 8:30–9:00 a.m. to minimize natural changes in corticosterone levels. After the rats were weighed and the caudal vein was catheterized, dexamethasone (0.2 mg/kg) was administered. Two hours later, the baseline blood sample was drawn. Rats ($n = 32$) were then randomly divided into the following four groups, with eight rats in each group: etomidate, ET-26-HCl, CPMM, or propylene glycol vehicle

control, and the infusion protocol started (see Fig. 1A). The infusion rate in each group was the corresponding predetermined MIR for that compound, and the drug infusion persisted for 60 min. After 30 min of drug infusion, adrenocorticotropic hormone (ACTH) was administered intravenously, and this was repeated every 30 min. The second blood sample was collected at the end of the infusion, and then blood samples were drawn every 30 min for 3.5 h. Blood samples (approximately 0.2 mL each) were kept at room temperature for 10 to 60 min, until coagulation, and then centrifuged at 3,500 gravity ($g$) for 5 min. The supernatant was transferred to a clean Eppendorf centrifuge tube and subjected to a second centrifugation (3,500 rpm for 5 min). After the second centrifugation, the supernatant was removed again, and a final high-speed centrifugation was performed to remove all red blood cells and particulate. The serum was transferred to a clean tube and immediately placed at −20 °C. The corticosterone concentration in each serum sample obtained at the various time points was quantified within 1–2 days using an enzyme-linked immunosorbent assay (ELISA kit, R & D Systems) and a 96-well plate reader.

During all studies, rats were placed on a warming stage, rectal temperatures were maintained at 36–38 °C, and oxygen was continuously provided at 2 L/min.

## Statistical analysis

All data are presented as the mean ± standard deviation. For comparisons of serum corticosterone concentrations after infusion of vehicle and test compounds, a one-way analysis of variance was conducted followed by Tukey's honest significant difference test or the Tamhane test. A value of $P < 0.05$ was considered statistically significant. All statistical analyses were carried out using the Statistical Package for Social Sciences version 21.0 software (Chicago, IL, USA). Figure preparation and curve fitting was performed with Prism version 5.0 (GraphPad Software, Inc., La Jolla, CA, USA).

## RESULTS

### Minimum infusion rate of each drug

The MIRs of etomidate, CPMM, and ET-26-HCl were 0.29, 0.95, and 0.62 mg/kg/min, respectively (Table 1).

### Evaluation of adrenocortical function

The baseline concentrations of serum corticosterone (before the start of the drug infusion) were not significantly different among the four groups and averaged 185.86 ± 68.66 ng/mL. However, serum corticosterone concentrations increased over time in all groups after drug infusion and ACTH stimulation. Compared with the control group, rats administered etomidate demonstrated significantly lower corticosterone concentrations at 60 min, which is the end of the drug infusion, as well as 90, 120, 150, and 240 min after the beginning of drug infusion. Compared with the etomidate group, rats administered ET-26-HCl showed significant differences 90, 120, 180, 210, and 240 min after begin of drug infusion. Compared with rats administered CPMM, rats administered etomidate showed significant differences in serum corticosterone concentrations at the end of the drug infusion as well as 60, 90, 180, and 210 min after begin of drug infusion. ACTH-stimulated

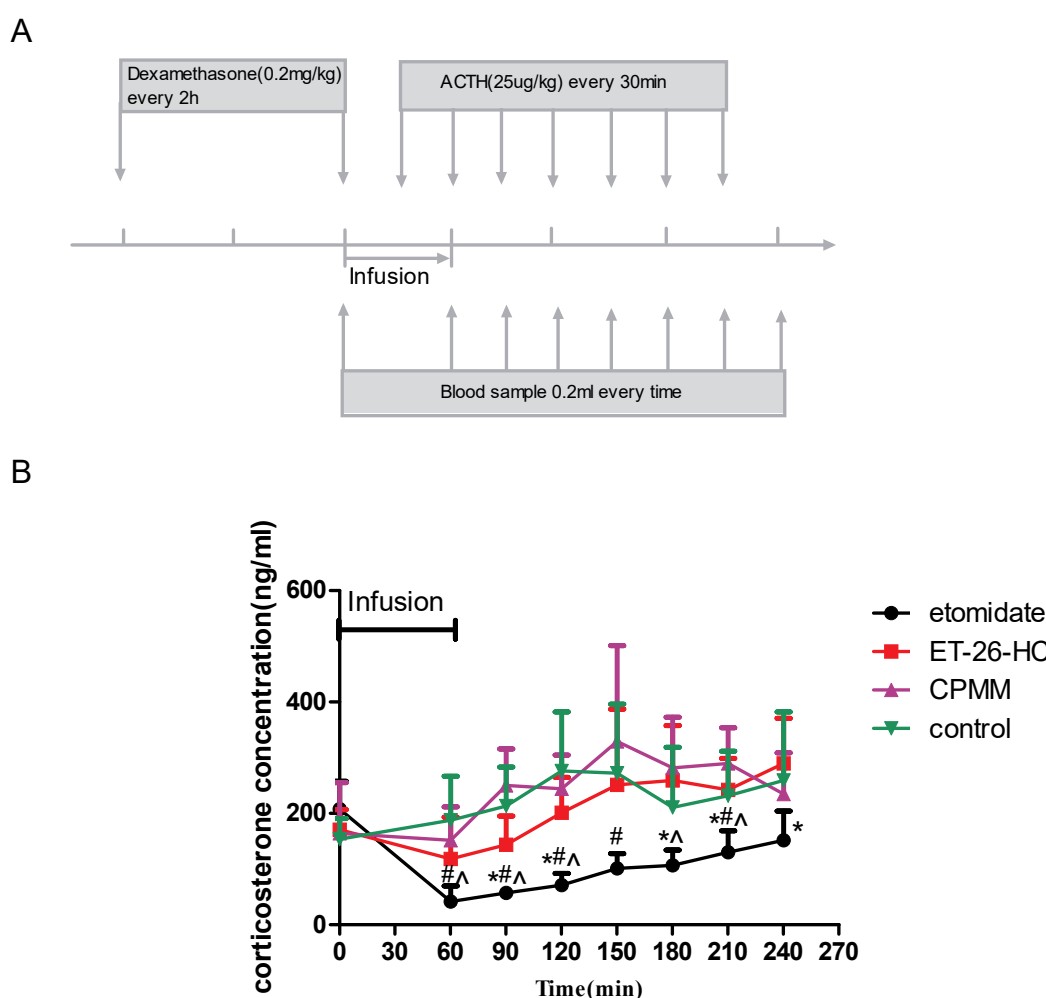

**Figure 1 Determination of serum corticosterone concentrations.** (A) Schematic depicting the experimental protocol. Before the hypnotic drug infusion, the first blood sample was drawn as the baseline. Adrenocorticotropic hormone (ACTH) was injected intravenously after 30 min of drug infusion and then once every 30 min for the duration of the experiment. The second blood sample was collected at the end of the drug infusion, and then blood samples were drawn every 30 min for 3.5 h. (B) Adrenocortical function as determined by serum corticosterone concentrations after hypnotic drug infusion. $^*P < 0.05$, for etomidate versus ET-26-HCl; $^\#P < 0.05$ for etomidate versus control; $\hat{P} < 0.05$ for etomidate versus CPMM. Eight rats were used in each group.

serum corticosterone concentrations in rats administered CPMM or ET-26-HCl were not significantly different from each other at any time (Fig. 1B).

## DISCUSSION

In the present study, we evaluated the MIRs of etomidate, ET-26-HCl, and CPMM by recording in anesthetized rats either a positive (+) or no (−) reaction to a painful stimulus, an up-and-down design method previously described (*Li et al., 2012*). Each hypnotic was administered intravenously for 40 min to determine its MIR because our preliminary study showed that the half-life of ET-26-HCl was 6–7 min, the longest of the three hypnotics

Table 1  Determination of the minimum infusion rate.

| Etomidate | | | ET-26-HCl | | | CPMM | | |
|---|---|---|---|---|---|---|---|---|
| IR | Result | MI | IR | Result | MI | IR | Result | MI |
| 0.33 | + | | 0.65 | + | | 1 | + | |
| 0.30 | + | | 0.59 | − | 0.62 | 0.9 | − | 0.95 |
| 0.27 | − | 0.285 | 0.65 | + | | 1 | + | |
| 0.30 | + | | 0.59 | − | 0.62 | 0.9 | − | 0.95 |
| 0.27 | − | 0.285 | 0.65 | − | | 1 | + | |
| 0.30 | + | | 0.72 | + | | 0.9 | − | 0.95 |
| 0.27 | − | 0.285 | 0.65 | + | | 1 | + | |
| 0.30 | + | | 0.59 | − | 0.62 | 0.9 | − | 0.95 |
| 0.27 | − | 0.285 | 0.65 | + | | 1 | + | |
| 0.30 | + | | 0.59 | − | 0.62 | 0.9 | − | 0.95 |
| 0.27 | − | 0.285 | 0.65 | + | | | | |
| | | | 0.59 | − | 0.62 | | | |
| MIR = 0.285 | | | MIR = 0.6 | | | MIR = 0.95 | | |

Notes.

A change in the response from negative to positive or positive to negative was defined as a pair, and the stimulation was repeated at different infusion rates until five pairs of responses were recorded. The minimum infusion rate was determined as the average of these five mean values.

IR, the infusion rate of each rat (mg/kg × min); MI, mean infusion rate for a pair of responses (mg/kg × min); MIR, minimum infusion rate (mg/kg × min).

used in the present study, and because it is generally acknowledged that the *in vivo* plasma concentration of drugs continuously infused at a constant rate reaches equilibrium at 4–5 half-lives. We determined that the MIR for etomidate was 0.285 mg/kg/min, for ET-26-HCl it was 0.62 mg/kg/min, and for CPMM it was 0.95 mg/kg/min. These results suggested that the anesthetic efficacy of ET-26-HCl was approximately one-half to one-third of that for etomidate, which is consistent with the results of our previous study (*Yang et al., 2017*). In addition, the MIR of CPMM found in the present study was consistent with the results of Ge and colleagues, which suggested that the immobilizing $ED_{50}$ (effective dose for 50 percent of the group) of CPMM is $0.89 \pm 0.18$ mg/kg/min (*Ge et al., 2012*). These authors also found that the total doses of etomidate and CPMM needed in a 2-h closed-loop infusion protocol to maintain an 80% electroencephalographic burst suppression ratio are 36 mg/kg and 143 mg/kg, respectively, indicating that the average infusion rates for these hypnotics are 0.3 mg/kg/min and 1.19 mg/kg/min, respectively. This infusion rate for etomidate is consistent with the rate found in the present study, while the rate for CPMM in the present study is slightly lower than that observed in the previous study. Thus, we speculate that all rats in the three groups used in the present study were at a similar depth of anesthesia.

After determining the MIRs, we next examined the effects of etomidate, ET-26-HCl, and CPMM continuously infused for 1 h and inducing the same anesthesia depth on serum corticosterone concentrations. Compared with those in control rats, ACTH-stimulated serum corticosterone concentrations were significantly decreased by etomidate, while those following ET-26-HCl or CPMM administration were not associated with a

significant difference. When ACTH-stimulated serum corticosterone concentrations in rats administered etomidate were compared with those in rats infused with vehicle, ET-26-HCl or CPMM, all time points examined after the drug infusions, except 180 min and 240 min, show significant differences with etomidate. The corticosterone concentrations tended to be reduced for the first hour following ET-26-HCl infusion (as compared to vehicle infusion); however, this reduction did not reach statistical significance. We concluded that ET-26-HCl does not induce obvious inhibition of adrenal function.

The safe dosage range of etomidate has been diminishing owing to its inhibition of adrenocortical function, as the suppression following even a single bolus may last 72 h (*Molenaar et al., 2012*). This inhibition is mainly the result of the high-affinity binding between the basic nitrogen in the imidazole ring of etomidate and the heme iron on 11β-hydroxylase (*Den Brinker et al., 2008*; *Fellows et al., 1983*; *Shanmugasundararaj et al., 2013*). In the 1980s, etomidate was used as a sedative for critically ill patients; however, in 1983, Watt and colleagues found that the continuous infusion of etomidate may increase mortality, and they speculated that the increased mortality is mainly caused by adrenocortical suppression. A series of studies later verified this speculation and recommended not to blindly administer etomidate to critical patients (*Morris & McAllister, 2005*). However, no other anesthetic currently possesses the characteristics of etomidate, such as to the ability to maintain stable hemodynamics, especially in aged or critically ill patients. Therefore, researchers have devoted much effort to develop new etomidate analogs that preserve the advantages but reduce the disadvantages of etomidate. There are at least two ways to achieve this goal. The first method involves designing a series of analogs that are rapidly metabolized so that adrenocortical inhibition stops soon after the administration is discontinued. With this in mind, MOC-etomidate and CPMM were designed by researchers at the Massachusetts General Hospital. Among these compounds, CPMM showed the greatest promise for development (*Campagna et al., 2014*; *Cotten et al., 2010*; *Cotten et al., 2009*; *Pejo et al., 2012*; *Santer et al., 2015*). The second method involves changing the molecular structure of etomidate to minimize adrenocortical suppression. It is widely acknowledged that the primary mechanism of adrenocortical suppression is through the interaction between the basic nitrogen in the imidazole ring of etomidate and the heme iron of 11β-hydroxylase (*Gay et al., 2009*; *Ouellet, Podust & De Montellano, 2008*; *Roumen et al., 2007*). Several studies have shown that anesthesia efficacy is significantly decreased when the basic nitrogen is replaced with other chemical groups, such as in carboetomidate, which has an anesthesia potency approximately one-seventh of that for etomidate (*Cotten et al., 2010*). However, Atucha and colleagues suggested that the imidazole carboxylic acid ester side chain of etomidate affects both anesthetic potency and adrenocortical function (*Atucha et al., 2009*). The design of ET-26-HCl is based on modifications of this side chain, and our previous study showed that ET-26-HCl produces definite and reversible anesthesia in beagle dogs. In the present study, no significant difference was observed in the serum corticosterone concentration after the continuous infusion of ET-26-HCl or vehicle, suggesting that any adrenocortical suppression induced by ET-26-HCl would be lower than that caused by etomidate. These results also indicated that new analogs may be developed by means other than using soft analogs.

## CONCLUSION

The method to evaluate minimum infusion rate of present study is feasible, and it could maintain a similar anesthesia depth of all drugs. The corticosterone concentrations tended to be reduced for the first hour following ET-26-HCl infusion (as compared to vehicle infusion) was mainly because of the molecular structure of ET-26-HCl still have some depression to the release of corticosterone; however, this reduction did not reach statistical significance. Thus, further studies are warranted examining the practicability of using ET-26-HCl as an infused anesthetic.

**Abbreviations**

| | |
|---|---|
| **MIR** | minimum infusion rate |
| **ACTH** | adrenocorticotropic hormone |
| **ET-26-HCl** | methoxyethyletomidate hydrochloride |
| **CPMM** | cyclopropyl-methoxycarbonylmetomidate |

### Funding

This work was supported by National Science and Technology Major Project 2014ZX09101001003 and 2014ZX09101001004. The funders had no role in study design, data collection and analysis, decision to publish, or preparation of the manuscript.

### Grant Disclosures

The following grant information was disclosed by the authors:
National Science and Technology Major Project: 2014ZX09101001003, 2014ZX09101001004.

### Competing Interests

The authors declare there are no competing interests.

### Author Contributions

- Junli Jiang conceived and designed the experiments, performed the experiments, analyzed the data, wrote the paper, prepared figures and/or tables, reviewed drafts of the paper.
- Bin Wang conceived and designed the experiments, analyzed the data, reviewed drafts of the paper.
- Zhaoqiong Zhu and Jin Liu contributed reagents/materials/analysis tools, reviewed drafts of the paper.
- Jun Yang contributed reagents/materials/analysis tools, reviewed drafts of the paper, the design of ET-26-HCl.
- Wensheng Zhang conceived and designed the experiments, reviewed drafts of the paper.

### Animal Ethics

The following information was supplied relating to ethical approvals (i.e., approving body and any reference numbers):

All animal protocols used in the present study were approved by the Ethics Committee of the West China Hospital, Sichuan University, China (ethics approval No. 2015015A; date: 28/12/2012).

## Data Availability

Jiang, Junli (2017): New draft item. figshare.

https://doi.org/10.6084/m9.figshare.5139853.v1.

## Supplemental Information

Supplemental information for this article can be found online at http://dx.doi.org/10.7717/peerj.3693#supplemental-information.

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
