# Peer review of "Minimum infusion rate and adrenocortical function after continuous infusion of the novel etomidate analog ET-26-HCl in rats"

_PeerJ, doi:10.7717/peerj.3693_

## Round 0.1 · original submission · Major Revisions

· Academic Editor

Major Revisions

Both reviewers suggest "Major revisions" for your manuscript.

Please address the statistical issues raised by Reviewer 1.

I also suggest you update Conclusion part of the manuscript to present discussion for wider readers' benefit.

·

Basic reporting

This manuscript is well written. Figure 1 should be improved by changing the color of either the Control or CPMM data as they are both green and it is difficult to resolve one from the other because the datasets overlap at several time points.

Experimental design

The experimental design is generally good.

Validity of the findings

1. The authors conclude that ET-26-HCl does not significantly suppress adrenocortical function in rats. While there was no statistical difference between the ET-26-HCl and control groups, the average corticosterone concentration immediately following an ET-26-HCl infusion (i.e. at the 60 min time point) was about half that following the vehicle (control) infusion. Rats in the ET-26-HCl group continued to have lower corticosterone concentrations than those in the control group at the 90 min and 120 min time points. Given the large variability in corticosterone concentrations within each group, the modest number of rats in each group (8), and the multiple comparisons being statistically tested, I suspect that the power to detect a difference is probably low (i.e. not high enough to detect even a 50% reduction in corticosterone concentration). So while one can conclude that ET-26-HCl produces significantly less adrenocortical suppression than etomidate, I don't think that it is appropriate to simply say that it doesn't produce significant adrenocortical suppression. To be more complete and accurate, the authors should write (both in the abstract and discussion) that the corticosterone concentrations tended to be reduced for the first hour following ET-26-HCl infusion (as compared to vehicle infusion); however, this reduction did not reach statistical significance. Ideally, they would also be able to comment on their power to detect differences among groups in the methods section.

2. The purpose of giving dexamethasone prior to starting the experiment is to reduce endogenous ACTH production and, thus baseline corticosterone concentrations to a near zero value. But the baseline corticosterone concentrations measured at time 0 are relatively high, approximating the ACTH-stimulated values measured in the control rats. That seems like at odd result.

Additional comments

1. Line 202: Carboetomidate is not a rapidly metabolized etomidate analog. It is an analog designed not to suppress adrenocortical function (i.e. second method).

2. Line 175: The pharmacology of a drug is defined by its structure, not how its made. Two identical drugs made by different synthetic processes will have the same pharmacology unless another compound is also present (i.e. an impurity). So this sentence needs clarification.

Reviewer 2 ·

Basic reporting

1). The manuscript is clearly written. Figures are relevant and well described. However the label of the Figure 1B Y axis contains the misprint: concentretion should be changed to concentration
2) According the Materials and Methods section (line 110) the infusion protocol for each rat began at 8:30–9:00 a.m. However, according to the Table 1 in the Supplemental materials the CPMM experiment was started at 13:16:00. Besides, the authors gave the following description of the animals: Male Sprague Dawley rats weighing 225–250 grams (line 64). These rats were acclimatized for 1 week (line 69). Based on that it is not clear how these rats became more than 300 grams in the CPMM experiment (Supplemental data, Table 1). It looks like the data for the CPMM experiment were taken from another study, and if it is so, it should be explained in the Materials and Methods section.

Experimental design

1) The authors have to explain why the experiment for the minimum infusion rate determination was conducted during 40 minutes and the time of the following experiment for the evaluation of the suppression of adrenocortical function (based on the results of the minimum infusion rate determination) was changed up to 1 hour.
2) According to the data given in Table 1 MIR (minimum infusion rate) is equal to MI (mean infusion rate for a pair of responses). Authors have to clarify why the mean infusion rate for a pair of responses is taken as a minimum infusion rate and how they may be sure that this choice is good for the depth of anesthesia and animals will still have negative reaction to a painful stimulus.

Validity of the findings

One of the aims of the study was to evaluate adrenocortical function after a continuous infusion of ET-26-HCl. The Conclusion of the manuscript is: Unlike with etomidate, a 60-minute infusion of the novel etomidate analog ET-26-HCl did not significantly suppress adrenocortical function in rats. However, the results of the experiment (Fig.1B) demonstrate the high variability of the trait (corticosterone concentration) both in control and ET-26-HCl treated rats. The corticosterone concentration was about half lower in ET-26-HCl treated rats as compared to controls just after infusion of ET-26-HCl was finished (the point corresponds to 60 min in the Fig.1B) and 30 minutes later the corticosterone concentration in ET-26-HCl treated rats was about 30% lower than in the controls. As soon as there were only 8 rats in each group one may suggest that the same experiment conducted on the larger group of animals may show that ET-26-HCl suppress adrenocortical function in rats but probably not so dramatically as etomidate does.

Additional comments

This is a very interesting and potentially promising for clinical studies experiment. However, the work needs more statistics.

---

## Round 0.2 · Minor Revisions

· Academic Editor

Minor Revisions

After first round of reviews the manuscript has been improved, though there are still remaining very minor remarks.

·

Basic reporting

Grammatical comment: On page 11 line 19, the authors write "With this in mind, MOC-etomidate and CPMM were designed by Massachusetts General Hospital." This sentence should be corrected to read "With this in mind, MOC-etomidate and CPMM were designed by researchers at the Massachusetts General Hospital."

Experimental design

Acceptable

Validity of the findings

Acceptable

Reviewer 2 ·

Basic reporting

The manuscript was improved.
However, there are two more comments on the manuscript:
1) The conclusion of the manuscript was revised and now it exactly reflects the results described in the manuscript. However, the authors have to change the text of conclusion at the end of the manuscript, too. See lines 232-233 in the .pdf text.
2). The Fig. 1B was substantially revised. The new colors improved the figure. Besides the standard deviations look differently as compared to the previos version of the Figure and no explanation was given either in the text of Methods section or in Figure itself. Do the authors still show the standard deviations or something else in Fig. 1B?

Experimental design

no comments

Validity of the findings

OK as soon as the conclusion became more accurate and exactly reflects the results of the study.

Additional comments

no comments

---

## Round 0.3 · Minor Revisions

· Academic Editor

Minor Revisions

I found some typos in the text. Please take into account minor remarks from the file attached (changes are in PDF)

I recommend update first phrase in the abstract starting from etomidate function overall. Make it similar to the first sentences in the introduction.

There are some evident typos (no spaces between words), for example

Page 1 in the title... infusionof-> infusion of
Page 4 ahumidity -> a humidity

Please try update Conclusion section. It is too short. Why corticosterone concentration was reduced for the first hour following ET-26-HCl infusion? Does it depend on animal model or more complex phenomena? Add some phrases to discussion and conclusion parts of the manuscript about possible applications of the results for wider reader circle.

---

## Round 0.4 · accepted · Accept

· Academic Editor

Accept

Previous time we had some typos in the text. It is fixed now in the last reviewing round.